# Dynamic Contrast-Enhanced and Intravoxel Incoherent Motion MRI Biomarkers Are Correlated to Survival Outcome in Advanced Hepatocellular Carcinoma

**DOI:** 10.3390/diagnostics11081340

**Published:** 2021-07-26

**Authors:** Bang-Bin Chen, Yu-Yun Shao, Zhong-Zhe Lin, Chih-Hung Hsu, Ann-Lii Cheng, Chiun Hsu, Po-Chin Liang, Tiffany Ting-Fang Shih

**Affiliations:** 1Department of Medical Imaging, National Taiwan University Hospital, Taipei 100, Taiwan; bangbin@gmail.com (B.-B.C.); pochin.liang@gmail.com (P.-C.L.); 2Department of Radiology, College of Medicine, National Taiwan University, Taipei 100, Taiwan; 3Department of Oncology, National Taiwan University Hospital, Taipei 100, Taiwan; yuyunshao@gmail.com (Y.-Y.S.); zzlin7460@ntu.edu.tw (Z.-Z.L.); chihhunghsu@ntu.edu.tw (C.-H.H.); alcheng@ntu.edu.tw (A.-L.C.); hsuchiun@ntu.edu.tw (C.H.); 4Graduate Institute of Oncology, College of Medicine, National Taiwan University, Taipei 100, Taiwan; 5Department of Medical Oncology, National Taiwan University Cancer Center, Taipei 100, Taiwan; 6Department of Internal Medicine, National Taiwan University Hospital, Taipei 100, Taiwan; 7Department of Medical Imaging, National Taiwan University Hospital Hsin-Chu Branch, Hsin-Chu City 300, Taiwan

**Keywords:** magnetic resonance angiography, diffusion MRI, hepatocellular carcinoma, survival, lenalidomide

## Abstract

Objective: This study assessed dynamic contrast-enhanced (DCE)-MRI and intravoxel incoherent motion diffusion-weighted imaging (IVIM DWI) parameters to prospectively predict survival outcomes in participants with advanced hepatocellular carcinoma (HCC) who received lenalidomide, a dual antiangiogenic and immunomodulatory agent, as second-line therapy in a Phase II clinical trial. Materials and methods: Forty-four participants with advanced HCC who had progression after sorafenib as first-line treatment were prospectively enrolled. Pretreatment MRI parameters—obtained from DCE-MRI (peak, slope, AUC, K^trans^, K_ep_, and V_e_), apparent diffusion coefficient (ADC), and IVIM DWI (pure diffusion coefficient (D), pseudodiffusion coefficient (D*), and perfusion fraction (f))—were derived from the largest hepatic tumor. The Cox model was used to investigate the associations of the parameters with progression-free survival (PFS) and overall survival (OS). Results: Median PFS and OS were 2.3 and 8.0 months, respectively. Univariate analysis showed that participants with a high slope (*p* = 0.024), K_ep_ (*p* < 0.001), and ADC (*p* = 0.018) values had longer PFS than those with low values; participants with a small tumor size (*p* = 0.006), high slope (*p* = 0.01), ADC (*p* = 0.015), and f (*p* = 0.012) values had longer OS than those with low values did. Cox multivariable analysis revealed that K_ep_ (*p* < 0.001) and ADC (*p* = 0.009) remained independent predictors of PFS; slope (*p* = 0.003) and ADC (*p* = 0.009) remained independent predictors of OS. Moreover, K_ep_ and slope were still significant after Bonferroni correction was performed (*p* < 0.005). Conclusion: Both pretreatment DCE-MRI and IVIM DWI parameters, especially slope and ADC, may predict PFS and OS in participants with HCC receiving lenalidomide as second-line therapy.

## 1. Introduction

Hepatocellular carcinoma (HCC) is the third highest cause of cancer-related death, with an increasing global incidence. For advanced HCC, sorafenib is recommended as a first-line treatment according to the Barcelona Clinic Liver Cancer staging system [1]; however, its response rate is low, with a complete response rate of 0% and a partial response rate of only 2.2% in two pivotal Phase III trials [2,3,4]. Three tyrosine kinase inhibitors, namely, regorafenib, cabozantinib, and ramucirumab, and two checkpoint inhibitors, namely, nivolumab and pembrolizumab, were approved by the US Food and Drug Administration as second-line treatment options after prior sorafenib treatment [5]. Lenalidomide, which has both antiangiogenic and immunomodulatory effects, also demonstrated efficacy as a second-line treatment for advanced HCC [6,7]. Patients with advanced HCC generally have poor survival outcomes; therefore, before the administration of treatments, determining image biomarkers that can be used to identify patients who are likely to benefit from such treatments is imperative.

Dynamic contrast-enhanced (DCE) MRI is used to measure tissue perfusion, blood flow, and vascularity by analyzing a tissue signal-enhancement curve after contrast-agent administration [8]. It can be used to measure changes in tumor vascular permeability induced by antiangiogenic agents. In several clinical trials of new targeted therapies for HCC, DCE-MRI biomarkers using conventional gadolinium-based contrast agents have been used as early surrogates to predict clinical response and survival outcome [7,9,10,11]. For example, in patients who had received sorafenib plus metronomic tegafur/uracil therapy, the forward volume transfer constant (K^trans^) correlated well with tumor response and survival [10]. Another study found that high peak (difference between maximal and baseline signal intensity) reduction within one week was a favorable prognostic factor after systemic treatment [8]. However, vascular response determined by >40% K^trans^ reduction at 2 weeks did not correlate with treatment response after lenalidomide [7] and vandetanib [10] treatments.

Diffusion-weighted imaging (DWI), in which changes in the cellular density of tissue can be estimated on the basis of an apparent diffusion coefficient (ADC), showed considerable promise as an imaging biomarker in HCC [12,13]. The intravoxel incoherent motion (IVIM) model using multiple *b* values can be used to derive pseudodiffusion (D*), pure diffusion characteristics (D), and perfusion fraction (f). Studies showed that IVIM imaging biomarkers may differentiate histological grades of HCC [14,15] and predict prognosis in transarterial chemoembolization treatment [16]. For example, ADC and IVIM-derived D values exhibited high diagnostic performance in differentiating high-grade HCC from low-grade HCC [14]. ADC and Dslow ratios calculated at 24–48 h relative to baseline were reported to be independent predictors of response for HCC after transarterial chemoembolization [17]. However, according to our review of the literature, data concerning the combined use of DCE-MRI and IVIM DWI for predicting survival outcomes in patients with HCC are limited. Currently, no imaging biomarker is available to predict survival outcome in patients receiving second-line targeted therapy after first-line sorafenib treatment. 

To address the aforementioned literature gap, we conducted this study with the purpose of assessing DCE-MRI and IVIM DWI biomarkers to prospectively predict survival outcomes in participants with advanced HCC who had progression after first-line sorafenib and received lenalidomide as second-line therapy. We hypothesized that pretreatment MRI biomarkers obtained from DCE-MRI and IVIM DWI would predict the survival outcome before lenalidomide treatment. 

## 2. Materials and Methods

This prospective open-label, single-arm, single-center, investigator-initiated Phase II clinical trial, was approved by the Institutional Research Ethics Committee (protocol code: NTUH-REC No. 201105063MB, date of approval: 2 August 2011) of our institute (www.clinicaltrials.gov NCT01545804, Last accessed on 10 July 2021) [7]. Written informed consent was obtained from all the participants. No financial support from the industry was received. The authors had full control of the data and submitted information. 

### 2.1. Study Participants

The inclusion criteria were as follows: receiving a histological or clinical diagnosis of HCC, having documented progression under treatment with or intolerance to sorafenib or other systemic therapy, having an Eastern Cooperative Oncology Group score of 0 or 1, being classified into Child–Pugh class A, and having at least one measurable lesion according to Response Evaluation Criteria in Solid Tumors (RECIST) 1.1 [18]. The exclusion criteria are listed in Figure 1. Participants received lenalidomide (25 mg/day orally) on days 1–21 every 4 weeks. Tumor response was assessed according to RECIST 1.1 after 4 and 8 weeks of treatment and every 8 weeks thereafter. Participants were followed until death. The primary endpoint was 6-month progression-free survival (PFS) using RECIST 1.1. We collected clinical data, namely, age, sex, hepatitis status, cirrhosis, extrahepatic metastasis, macroscopic vascular invasion, alpha-fetoprotein, treatment received, and mortality date. 

We initially enrolled 55 participants during the period from June 2012 to June 2014. Lastly, we included 44 participants (men: 39; women: 5; median age: 60.1 ± 11.6 years; range 31–80 years) who had undergone a pretreatment MRI examination in this study (Figure 1). Data for all 44 participants were reported in our previous study [7], which focused on the correlation of a single DCE-MRI parameter (K^trans^) with treatment outcomes. In our previous study, the vascular response determined by a >40% decline in K^trans^ was not associated with any treatment outcome [7]. In contrast, in the present study, we investigated the correlation of pretreatment DCE-MRI and IVIM biomarkers with PFS and overall survival (OS), which had not been previously reported.

### 2.2. MRI Protocol

Each participant received liver MRI at 3 T (Magnetom Verio; Siemens Healthcare, Erlangen, Germany) through a 32-channel phased-array coil. Routine MRI sequences included the following: a half-Fourier single-shot turbo spin-echo sequence, a breath-hold T1-weighted dual-echo (inphase and opposed-phase) volumetric interpolated breath-hold examination sequence, and a T2-weighted fast spin-echo sequence with fat suppression. Three gradient directions were chosen for DWI (Table 1).

Axial liver IVIM imaging was performed using a free-breathing single-shot echo-planar imaging protocol in which diffusion gradients were applied in three orthogonal directions [19]. Encoding was performed using 16 *b* values (0, 10, 20, 30, 40, 50, 60, 70, 80, 90, 100, 200, 300, 400, 500, and 1000 s/mm^2^) before injection of gadolinium chelate. 

DCE-MRI was performed on 24 consecutive oblique coronal sections using a three-dimensional T1-weighted volumetric interpolated breath-hold examination (VIBE) sequence. Gadobutrol at a dose of 0.1 mmol/kg (Gd-DO3A-butrol, Gadovist; Bayer Pharma, Leverkusen, Germany) was injected at a rate of 2 mL/s into an antecubital vein by using an automated injector, followed by a 20 mL saline flush. All the participants were instructed to hold their breath for as long as they could tolerate, and then breathe slowly and smoothly during imaging. The total acquisition time for DCE-MRI was 2 min and 50 s, with a temporal resolution of 6.4 s; moreover, 600 dynamic images were obtained for each participant. Lastly, static fat-suppressed axial contrast-enhanced T1-weighted imaging was performed to image the whole liver. All imaging procedures were performed by the same technician.

### 2.3. Image Analysis

#### 2.3.1. IVIM Modeling of DWI

DWI data were coregistered using an image-based nonaffine registration algorithm (dynamic field correction), with a *b* value of 0 s/mm^2^ serving as a reference. For calculating the standard ADC value, monoexponential fitting of the data was calculated by a least-squares fit equation (S/S0 = exp (−b×ADC)) using all b values, where S0 is the signal without diffusion gradient, and S is the signal with a diffusion weighting [20,21]. Subsequently, IVIM parameters were derived with all *b* values serving as input data on a voxel-by-voxel basis. The following formula was used to derive the IVIM parameters [22]:S(*b*)/S(0) = f × exp[−*b*(D*)] + (1 − f) × exp[−*b*(D)
where D denotes a pure diffusion coefficient; D* denotes a pseudodiffusion coefficient; f denotes a perfusion fraction; and S(*b*) and S(0) denote signal intensity with and without the application of the diffusion gradient, respectively.

D values were estimated from signal-intensity data at high *b* values (*b*  >  200 s/mm^2^). 

Regions of interest (ROIs) were drawn using commercial Osirix^®^ medical-image software [23]. ADC and IVIM values were calculated using an Osirix plugin (ADCmap, version 2.4). For nonlinear least-squares fitting, the Levenberg–Marquardt algorithm was implemented in the plugin.

All image measurements were performed by a radiologist (B.B.C) with 12 years of experience in liver MRI interpretation; the radiologist was blinded to the clinical history of the participants. A single representative ROI was manually traced along the margin of the tumor on ADC maps on the section showing the largest tumor cross-sectional area. Subsequently, the ROI was copied and pasted to the images of IVIM parameters. The ROI location was visually checked to prevent misregistration due to motion. The mean (range) ROI area was 38.4 ± 44.4 (1.8–174.5) cm^2^ (Figure 2, Figure 3 and Figure 4). 

#### 2.3.2. DCE-MRI

DCE-MRI data were analyzed using a commercial software tool (MIStars; Apollo Medical Imaging, Melbourne, Australia), and motion correction was performed. The motion-correction algorithm used a 2D rigid body with three adjustable parameters: translation in x and y, and inplane rotation. The following semiquantitative parameters were obtained by analyzing the characteristics of tumor enhancement curves: Peak (maximal signal intensity minus baseline signal intensity), slope (maximal ascending slope of the curve), and initial area under the gadolinium concentration–time curve (AUC) at 60 s after contrast injection. Furthermore, pharmacokinetic modeling was calculated using a single-input two-compartment model with the aorta as arterial input function [9,24]. Three quantitative parameters (K^trans^: forward volume transfer constant, K_ep_: reverse rate transfer constant, and V_e_: extravascular extracellular space volume per unit volume of tissue) were automatically calculated pixel by pixel using a constrained nonlinear least-squares fitting algorithm with adjustable delay time. All ROIs were drawn by the same radiologist (B.B.C). The necrotic area within a tumor was included. The mean (range) ROI in the tumor was 49.5 ± 59.3 (2.1–288) cm^2^ (Figure 2 and Figure 3). 

In addition, to evaluate the interobserver variability of these parameters, ROI placement was performed for all MR imaging examinations in all patients by another radiologist (T.T.F.S., with 27 years of experience in MR imaging).

### 2.4. Statistical Analysis

Interobserver variability was calculated by using an intraclass correlation coefficient. Spearman’s correlation (rho) analysis was used to determine the correlation between MRI parameters (very weak correlation: <0.2; weak: 0.20–0.39; moderate: 0.40–0.59; strong: 0.60–0.79; very strong: 0.80–1.0). PFS and OS were measured from the date of examination to the date of progression and to the date of death, respectively. MRI parameters derived for participants with short and long OS (determined by a median OS period of 8.0 months) were compared using the nonparametric Mann–Whitney U test. Survival was analyzed using the Mantel–Cox log-rank test and was presented as Kaplan–Meier survival curves. For MRI parameters, the optimal cutoff for the predictor was estimated by using the maximally selected rank statistics (maxstat package) in R statistical software (R, version 4.0.3; R Foundation for Statistical Computing, Vienna, Austria). Age, sex, tumor size, alpha-fetoprotein, Eastern Cooperative Oncology Group performance status, Child–Pugh score, cirrhosis, macroscopic vascular invasion, extrahepatic spread, and MRI parameters were included in univariate Cox proportional-hazards regression models for PFS and OS. Variables with *p* values of <0.05 in the univariate analysis were used as inputs for a multivariable model. All the statistical analyses were performed using the R software and SPSS for Windows 22 (SPSS, Chicago, IL, USA). *p* < 0.05 was considered to indicate a significant difference. Bonferroni correction was applied to adjust for multiple comparisons (*p* < 0.005).

## 3. Results

### 3.1. Participants’ Characteristics

Mean tumor size was 38.4 ± 44.4 cm^2^ (range, 1.8–174.5cm^2^; median, 17.5cm^2^). Underlying liver diseases included hepatitis B (29/44, 66%), hepatitis C (8/44, 18%), and alcoholic liver disease (5/44, 11%) (Table 2). Among the 44 participants, the best RECIST responses were a partial response in 6 (14%) participants, stable disease in 18 (41%) participants, progressive disease in 19 (43%) participants, and a none-valuable response in 1 (2%) participant. All participants had died by December 2018. The 3-month and 6-month PFS rates were 48% (21/44) and 11% (5/44), respectively. Median PFS and OS were 2.3 (range, 0.8–16.8) and 8.0 (range, 1–54) months, respectively. 

### 3.2. Correlation of Tumor Size, DCE-MRI, and IVIM Parameters

The data of MRI parameters are shown in Table 3. Tumor size showed moderate inverse correlations with ADC (rho = −0.54, *p* < 0.001) and f (rho = −0.44, *p* = 0.001). D* showed weak correlations with K^trans^ (rho = 0.26, *p* = 0.045) and K_ep_ (rho = 0.26, *p* = 0.043). ADC and D were moderately correlated with each other (rho = 0.41, *p* = 0.005).

### 3.3. Intraclass Correlation Coefficients of MR Quantitative Parameters

Intraclass correlation coefficients for interobserver variability were 1.00 (95% CI: 1.00, 0.999) for peak, 0.999 (95% CI: 0.998, 0.999) for slope, 0.999 (95% CI: 0.998, 0.999) for AUC, 0.998 (95% CI: 0.996, 0.999) for K^trans^, 0.996 (95% CI: 0.993,0.998) for K_ep_, 0.992 (95% CI: 0.985, 0.996) for V_e_, 0.935 (95% CI: 0.883, 0.964) for ADC, 0.927 (95% CI: 0.860, 0.961) for D, 0.943 (95% CI: 0.799, 0.977) for D*, and 0.975 (95% CI: 0.944, 0.987) for f.

### 3.4. Comparison of MRI Parameters Derived for Participants with Short (≤8 Months) and Long (>8 Months) OS

Both ADC (*p* = 0.02) and f (*p* = 0.02) were significantly higher in participants with a long OS than in those with a short OS. However, both parameters were not significant after Bonferroni correction was performed. DCE-MRI parameters did not significantly differ between participants in these two subgroups (Table 4). 

### 3.5. Correlation of MRI Parameters with PFS and OS

Univariate analysis revealed that participants with a high slope (*p* = 0.024), K_ep_ (*p* < 0.0001), and ADC (*p* = 0.018) values had longer PFS than those with low values did (Table 5, Figure 5A–C). Large tumor size was a prognostic factor for poor OS (*p* = 0.006). Furthermore, participants with a high slope (*p* = 0.01), ADC (*p* = 0.015), and f (*p* = 0.012) values had longer OS than those with low values (Table 5, Figure 5D–F). After incorporating significant variables, Cox multivariable analysis revealed that K_ep_ (hazard ratio = 0.2; 95% confidence interval = 0.1, 0.5; *p* < 0.001) and ADC (hazard ratio = 0.3; 95% confidence interval = 0.1, 0.7; *p* = 0.009) remained independent predictors of PFS; slope (hazard ratio = 0.3; 95% confidence interval = 0.2, 0.7; *p* = 0.003) and ADC (hazard ratio = 0.3; 95% confidence interval = 0.1, 0.8; *p* = 0.009) remained independent predictors of OS. Moreover, K_ep_ and slope were still significant after Bonferroni correction was performed. 

## 4. Discussion 

Our study demonstrated that, in participants with advanced HCC who received lenalidomide as second-line therapy, high baseline slope, K_ep_, and ADC values were associated with better PFS, while small tumor size, high baseline slope, ADC, and f values were associated with better OS. Moreover, K_ep_ and ADC remained independent predictors of PFS, and slope and ADC remained independent predictors of OS in multivariable analysis, after adjusting clinical factors and tumor size. 

DCE-MRI provides both semiquantitative (peak, slope, and AUC) and quantitative (K^trans^, K_ep_, V_e_) parameters for assessing tumor angiogenesis [8]. Previous studies reported that high baseline peak and an early decrease in peak within 1 week were associated with more favorable OS in participants with HCC receiving antiangiogenic therapy [9,24]. In contrast, our study demonstrated that a high baseline slope value was associated with more favorable PFS and OS. Another study also revealed that high baseline peak and slope values derived for a tumor before radiotherapy were strongly correlated with a more favorable RECIST-based response rate [25]. Slope is closely related to tumor blood flow, whereas peak represents a complex process involving perfusion, blood flow, and vascular permeability within tumors [24]. These semiquantitative parameters have potential for predicting participants’ prognosis and have the advantage of easy calculation in daily practice; nevertheless, they may not be reproducible for different MRI machines or contrast injection flow rates. Therefore, standard and consistent MRI protocols are necessary for evaluating a treatment response or comparing results in multicenter clinical trials. 

DWI enables the quantitative assessment of tumor cellularity according to ADC values without the use of contrast agents; therefore, it is particularly useful in participants with severe renal dysfunction who are at risk of nephrogenic systemic fibrosis [12]. A retrospective study involving 58 patients with HCC who underwent drug-eluting embolic chemoembolization or radioembolization found that a lower ADC value was associated with a poorer PFS (*p* = 0.02) [13]. Another study used DCE-MRI and DWI in 20 patients with locally advanced HCC who had no treatment before concurrent chemoradiotherapy; the study revealed that patients with higher ADC values had significantly longer PFS than those with lower ADC values [26]. These findings are consistent with our results that a high pretreatment ADC value, representing low tumor cellularity or high tumor necrosis, was associated with more favorable survival outcomes. 

Regarding IVIM DWI, f is believed to represent the fractional blood volume of microcirculation. f is also influenced by other bulk flow phenomena, including glandular secretion and the blood flow pattern [27]. Previous studies demonstrated a significant correlation between f and the percentage of arterial enhancement [28] or enhancement ratios [20] in HCC. Therefore, f may represent the hypervascular portion of a tumor. Furthermore, f was significantly correlated with treatment response to sorafenib in HCC, and increased f after treatment suggested longer OS [29]. Although slope and f are both related to microcirculations in a tumor, we did not find a significant correlation between these two parameters. Therefore, slope and f may reflect different pathophysiologies of the tumor microenvironment in advanced HCCs. 

We identified a weak positive correlation of D* with K^trans^ (rho = 0.26, *p* = 0.045) and K_ep_ (rho = 0.26, *p* = 0.043). D* represents the length and velocity of the capillary network in a tumor [27], whereas K^trans^ and K_ep_ both represent vascular permeability. K^trans^ and K_ep_ were reported to predict treatment response or survival outcome in patients with HCC receiving antiangiogenic therapy [11,30]. In this study, we also found that high baseline K_ep_ was associated with better PFS, but not OS.

Although ADC and D were moderately correlated with each other (rho = 0.41, *p* = 0.005), we did not find the correlation of D with clinical outcomes. ADC incorporates the information of tumor perfusion (low *b* values) and cellularity (high *b* values). It seems that tumor perfusion, rather than cellularity, was more likely to predict survival outcomes in our study population, probably due to the hypervascular characteristics of HCC and antiangiogenic effect of lenalidomide. Similarly, slope and K_ep_ were also related to tumor perfusion, and both were independent predictors for survival outcomes.

Our study has several limitations. First, our sample size was small, and all patients were enrolled from a single institute. Second, all MRI parameters were measured on a single slice containing the largest tumor cross-sectional area instead of the whole tumor. Third, the acquisition time for IVIM was relatively long because multiple *b* values were obtained [31]. A previous study found that D and f calculated by the simplified IVIM model from three *b* values provided more discriminatory power between liver lesions than ADC determined from two *b* values did [32]. Fourth, the use of least-squares fitting can introduce biases for the estimation of f and D*, and this can be improved using recent Bayesian approaches because they allow for the use of prior information to regularize the fitting and to introduce a spatial dependency between voxels [33,34]. Future study is necessary to compare the goodness-of-fit between the biexponential IVIM model and recent Bayesian approaches. Lastly, because DCE-MRI was performed on a coronal view, and T1WI/DWI/IVIM were performed on an axial view, we did not coregister these images. In a future study, we aim to produce these images on an axial view for better visualization.

In conclusion, both pretreatment DCE-MRI and IVIM DWI biomarkers, especially slope and ADC, may predict survival outcomes in participants receiving lenalidomide as second-line therapy. These biomarkers provide information that transcends mere morphology and have potential for use in pretreatment selection of participants who are likely to benefit from second-line targeted therapy.

## Figures and Tables

**Figure 1 diagnostics-11-01340-f001:**
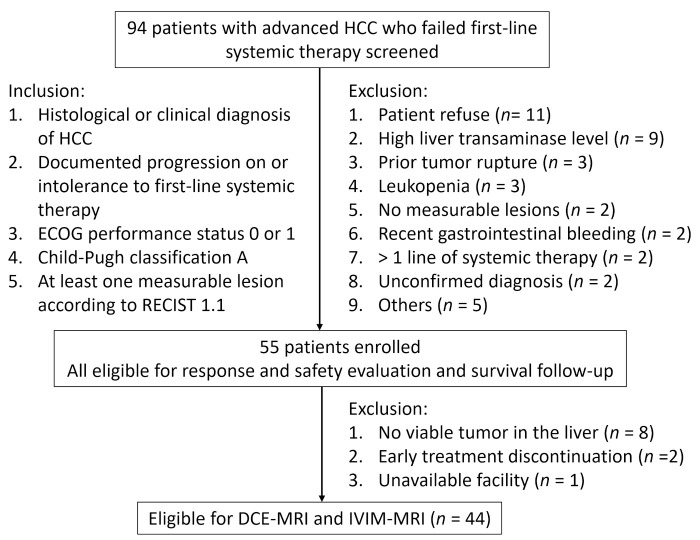
Summary of inclusion and exclusion criteria, and final study population. HCC = hepatocellular carcinoma; ECOG = Eastern Cooperative Oncology Group; RECIST = response evaluation criteria in solid tumors; DCE = dynamic contrast-enhanced; IVIM = intravoxel incoherent motion.

**Figure 2 diagnostics-11-01340-f002:**
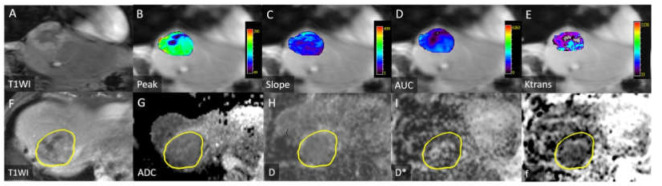
Images for a 77-year-old man with progression-free survival of 3.6 months and overall survival of 10 months. (**A**) Coronal contrast-enhanced T1-weighted image depicting a hepatocellular carcinoma in the right liver dome. Peak (**B**), slope (**C**), AUC (**D**), and K^trans^ (**E**) values were 15.4 (%), 12.8 (1/s), 1066 (/10 s), and 0.079 (min^−1^), respectively. (**F**) Axial contrast-enhanced T1-weighted images depicting heterogeneous enhancement of the tumor. ADC (**G**), D (**H**), D* (**I**), and f (**J**) values were 1.38 × 10^−3^, 1.16 × 10^−3^, and 100 × 10^−3^ s/mm^2^, and 11.7%, respectively. Yellow circles in Figure 2F–J mark tumor margin.

**Figure 3 diagnostics-11-01340-f003:**
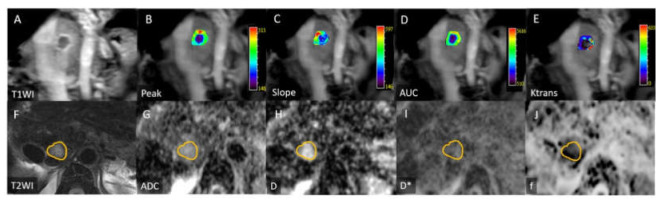
Images for a 71-year-old woman with progression-free survival of 5.5 months and overall survival of 29.8 months. (**A**) Coronal contrast-enhanced T1-weighted image depicting a hepatocellular carcinoma with peripheral enhancement in the right liver dome. Peak (**B**), slope (**C**), AUC (**D**), and K^trans^ (**E**) values were 36.2 (%), 38.4 (1/s), 3552 (/10 s), and 0.119 (min^−1^), respectively. (**F**) Axial T2-weighted image depicting mild hyperintense signal intensity of the tumor. ADC (**G**), D (**H**), D* (**I**), and f (**J**) values were 1.96 × 10^−3^, 1.6 × 10^−3^, and 100 × 10^−3^ s/mm^2^, and 16.7 (%), respectively. Yellow circles in Figure 3F–J mark tumor margin.

**Figure 4 diagnostics-11-01340-f004:**
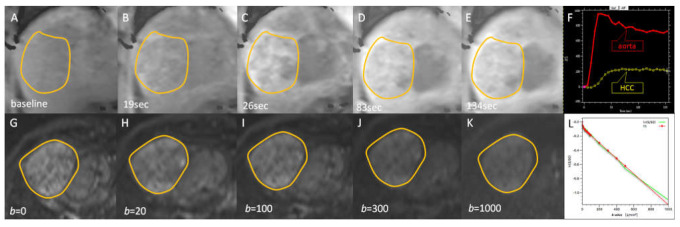
Representative signal dynamics images for DCE-MRI and DWI/IVIM in a 53-year-old man. (**A**–**E**) Coronal contrast-enhanced T1-weighted image depicting dynamic enhancement of a hepatocellular carcinoma (HCC) in right liver dome. (**F**) DCE-MRI enhancement curves of aorta (red) and HCC (yellow). (**G**–**K**) Axial DWI/IVIM images with different b values. (**L**) Fitting diffusion curve in whole HCC lesion. Yellow circles in Figure 4F–J,G–K mark tumor margin.

**Figure 5 diagnostics-11-01340-f005:**
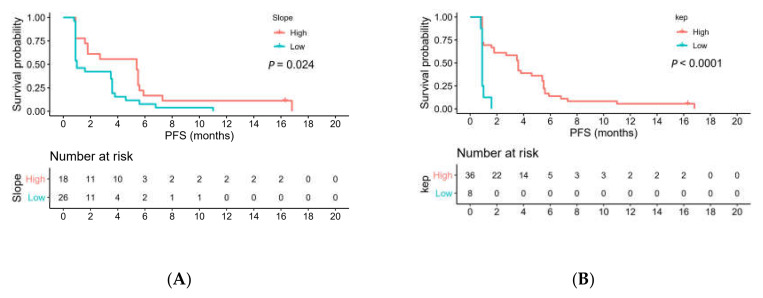
Kaplan–Meier curves indicating that participants with high pretreatment (**A**) slope, (**B**) K_ep_, and (**C**) ADC values had longer PFS than those with low values did. High pretreatment (**D**) slope, (**E**) ADC, and (**F**) f values had a longer OS than those with low values did. Cutoffs for MRI biomarkers were determined by using maximally selected rank statistics (maxstat package) in R statistical software.

**Table 1 diagnostics-11-01340-t001:** MRI parameters.

Sequence	TR (msec)	TE (msec)	Flip Angle (degrees)	Matrix	Field of View (mm)	Slice Thickness/Gap (mm)	NEX	Acquisition Time (min)
Coronal HASTE	1400	93	160	640 × 640	320 × 320	5/0	1	0:59
Axial T1WI VIBE (in- and opposed-phase)	4.2	2.5	10	512 × 416	350 × 284	3/0	1	0:52
Axial T2WI FS	2610	96	123	640 × 440	340 × 234	6/0	1	1:04
DWI (*b* = 50, 500, 1000)	7300	83	90	384 × 300	4000 × 313	6/0	2	4:08
IVIM (*b* = 0, 10, 20, 30, 40, 50, 60, 70, 80, 90, 100, 200, 300, 400, 500, 1000)	4438	77.2	180	182 × 150	385 × 317	6/0	2	14:06
DCE-MRI	4.2	2.5	9	420 × 448	400 × 313	5/0	1	2:50 (25 sets)
Axial T1WI postcontrast	141	2.5	70	320 × 220	340 × 234	6/0	1	1:00

Note: HASTE = half-Fourier single-shot turbo spin-echo; T1WI = T1-weighted imaging; VIBE = volumetric interpolated breath-hold examination; T2WI FS = T2-weighted imaging with fat suppression; DWI = diffusion-weighted imaging; IVIM = intravoxel incoherent motion; DCE = dynamic contrast-enhanced; TR = repetition time; TE = echo time; NEX = number of averages.

**Table 2 diagnostics-11-01340-t002:** Clinical characteristics and MRI parameters of 44 participants.

Characteristic	Value
Age (y)	60.0 ± 11.6
Sex (men/women)	39/5
Size (cm^2^)	38.4 ± 44.4
ECOG (0/1)	8/36
Child–Pugh score (5/6)	23/21
HBsAg (+)	29 (66)
Anti-HCV (+)	8 (18)
Alcohol abuse ^†^	5 (11)
Cirrhosis ^††^	34 (77)
Extrahepatic metastasis	38 (86)
Macroscopic vascular invasion	24 (55)
Serum AFP > 400 ng/mL	29 (66)
Prior treatment	
Surgery	20 (45)
Ablation	8 (18)
TACE	35 (80)
Sorafenib	44 (100)

Note: Unless otherwise indicated, data are mean ± standard deviation, and data in parentheses are percentages. AFP = alpha-fetoprotein; TACE = transarterial chemoembolization; ECOG = Eastern Cooperative Oncology Group; HBsAg = hepatitis B surface antigen; Anti-HCV = antihepatitis C antibody. ^†^ Alcohol abuse was defined as history of more than 3 drinks a day, documentation of alcoholism or alcohol abuse in a physician’s progress notes, or a history of alcoholic hepatitis. ^††^ Cirrhosis was graded histologically (*n* = 4) or clinically (combined laboratory data and imaging, *n* = 30).

**Table 3 diagnostics-11-01340-t003:** MRI parameters of 44 participants.

MRI Parameters	
Peak (%)	27.5 ± 10.5
Slope (1/s)	19.5 ± 7.8
AUC (/10 s)	2635 ± 1101
K^trans^ (min^−1^/1000)	152 ± 146
K_ep_ (min^−1^/1000)	1252 ± 1248
V_e_ (%)	10.7 ± 6.2
ADC (10^−3^ mm^2^/s)	1.4 ± 0.3
D (10^−3^ mm^2^/s)	1.1 ± 0.3
D* (10^−3^ mm^2^/s)	65.1 ± 38
f (%)	17.6 ± 10.6

Note: AUC = area under the curve; K^trans^ = forward volume transfer constant; K_ep_ = reverse rate transfer constant; V_e_ = extravascular extracellular space volume per unit volume of tissue; ADC = apparent diffusion coefficient; D = pure diffusion coefficient; D* = pseudodiffusion coefficient; f = perfusion fraction.

**Table 4 diagnostics-11-01340-t004:** Comparison of MR parameters in participants with short (≤8 months) and long (>8 months) overall survival.

Parameters	Short (*n* = 23)	Long (*n* = 21)	*p* Value
Peak (%)	270 ± 101	280 ± 112	0.85
Slope (1/s)	18.8 ± 7.2	20.4 ± 8.6	0.45
AUC (/10 s)	2660 ± 1031	2607 ± 1199	0.83
K^trans^ (min^−1^/1000)	106 ± 75	203 ± 186	0.08
K_ep_ (min^−1^/1000)	943 ± 647	1591 ± 1630	0.10
V_e_ (%)	9.6 ± 5.4	11.9 ± 7	0.36
ADC (10^−3^ mm^2^/s)	1.28 ± 0.2	1.54 ± 0.4	0.02 *
D (10^−3^ mm^2^/s)	1.06 ± 0.2	1.14 ± 0.4	0.93
D* (10^−3^ mm^2^/s)	65.6 ± 38.8	64.5 ± 38	0.87
f (%)	14.6 ± 9.1	20.8 ± 11.5	0.02 *

Note: Data are mean ± standard deviation. AUC = area under the curve; K^trans^ = forward volume transfer constant; K_ep_ = reverse rate transfer constant; V_e_ = extravascular extracellular space volume per unit volume of tissue; ADC = apparent diffusion coefficient; D = pure diffusion coefficient; D* = pseudodiffusion coefficient; f = perfusion fraction. * *p* value indicates a significant difference. Employed statistical analysis was the Mann–Whitney test. Short and long OS were determined by a median OS period of 8.0 months.

**Table 5 diagnostics-11-01340-t005:** Univariate and multivariable Cox regression analysis for progression-free and overall survival.

Parameters		Progression-Free Survival (PFS)				Overall Survival (OS)			
		Univariate		Multivariable		Univariate		Multivariable	
	Cutoff	Hazard Ratio	*p* Value	Hazard Ratio	*p* Value	Hazard Ratio	*p* Value	Hazard Ratio	*p* Value
Age (y)	60	1.0 (0.5–1.8)	>0.99			1.0 (0.6–1.9)	0.89		
sex	M vs. F	0.5 (0.2–1.3)	0.18			1.0 (0.4–2.4)	0.98		
Size (cm^2^)	17.5	0.9 (0.5–1.7)	0.82			0.4 (0.2–0.8)	0.006 *		
ECOG	0 vs. 1	0.9 (0.4–2.1)	0.89			1.0 (0.4–2.1)	0.93		
Child	5 vs. 6	1.1 (0.6–2.0)	0.74			1.2 (0.7–2.3)	0.48		
AFP (ng/mL)	400	0.9 (0.5–1.7)	0.78			0.8 (0.4–1.5)	0.49		
Cirrhosis	No vs. yes	0.7 (0.4–1.5)	0.39			0.9 (0.4–1.8)	0.71		
Macroscopic vascular invasion	No vs. yes	0.99 (0.5–1.8)	0.98			1.8 (1.0–3.4)	0.06		
Extrahepatic spread	No vs. yes	1.1 (0.4–2.8)	0.89			1.0 (0.4–2.7)	0.97		
	PFS/OS								
Peak (%)	32.5/16.8	0.7 (0.4–1.4)	0.35			0.6 (0.3–1.4)	0.23		
Slope (1/s)	21.4/22.4	0.5 (0.3–0.9)	0.024 *	0.6 (0.3–1.1)	0.11	0.4 (0.2–0.8)	0.01 *	0.3 (0.2–0.7)	0.003 *
AUC (/10 s)	1115/3689	0.5 (0.2–1.3)	0.15			1.7 (0.8–3.7)	0.19		
K^trans^ (min^−1^/1000)	51/21	0.8 (0.4–1.5)	0.47			1.7 (0.5–5.5)	0.42		
K_ep_ (min^−1^/1000)	370/1730	0.2 (0.1–0.5)	<0.001 *	0.2 (0.1–0.5)	<0.001 *	0.7 (0.3–1.5)	0.3		
V_e_ (%)	6.5/11.1	0.7 (0.4–1.4)	0.34			1.6 (0.9–3.0)	0.11		
ADC (10^−3^ mm^2^/s)	0.943/1.138	0.3 (0.1–0.7)	0.018 *	0.3 (0.1–0.7)	0.009 *	0.5 (0.2–0.9)	0.015 *	0.3 (0.1–0.8)	0.009 *
D (10^−3^ mm^2^/s)	1.183/1.173	0.6 (0.2–1.3)	0.2			0.6 (0.3–1.3)	0.18		
D* (10^−3^ mm^2^/s)	10.9/10.9	2.5 (0.8–8.4)	0.1			3.1 (0.9–10.3)	0.051		
f (%)	28/23.4	0.6 (0.2–1.4)	0.17			0.4 (0.2–0.8)	0.012 *	0.6 (0.3–1.1)	0.1

Note—Data in parentheses are 95% confidence intervals. AUC = area under the curve; K^trans^ = forward volume transfer constant; K_ep_ = reverse rate transfer constant; V_e_ = extravascular extracellular space volume per unit volume of tissue; ADC = apparent diffusion coefficient; D = pure diffusion coefficient; D* = pseudodiffusion coefficient; f = perfusion fraction. * *p* value indicates significant difference as determined by Cox regression analysis.

## Data Availability

The datasets generated during and/or analysed during the current study are available from the corresponding author upon reasonable request.

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
