# Peer review of "Dynamic Contrast-Enhanced and Intravoxel Incoherent Motion MRI Biomarkers Are Correlated to Survival Outcome in Advanced Hepatocellular Carcinoma"

_diagnostics, 2021, doi:10.3390/diagnostics11081340_

Round 1

Reviewer 1 Report

Dear Authors, 

the paper "Dynamic contrast-enhanced and intravoxel incoherent motion MRI biomarkers are correlated survival outcome in advanced hepatocellular carcinoma" is well written and complete.

For me it may be suitable for publication in the present form.

Best Regards

Author Response

Thank you for the positive comments.

Reviewer 2 Report

General Comments

The paper aims at assessing if MRI biomarkers, extracted from Diffusion-Weighted Imaging - specifically Intravoxel-Incoherent Motion - and Dynamic Contrast-Enhanced sequence, might be correlated to survival outcomes in patients affected by hepatocellular carcinoma.

The study enrolled prospectively 44 patients who had progression after sorafenib as 1st-line treatment. Patients underwent MRI sessions before the start of the treatment with lenalidomide as 2nd-line therapy in a phase II trial.

The paper shows that some MRI biomarkers – especially ADC and Slope – are predictive of survival. This result is not new but the paper is interesting and scientifically sound. However, some changes must be done to improve the document.

The language must be improved throughout the document.

Specific Comments

Title

… MRI biomarkers are correlated survival outcome… should be modified as follow: MRI biomarkers are correlated to survival outcome.

Abstract

The objective must be rearranged. There is something missing in the sentence.

…Univariable analysis… should be modified as follows: univariate analysis

ADC is not a parameter that can be calculated from the IVIM biexponential model but can be surely generated using the mono-exponential model from DWI. Please clarify this aspect.

Introduction

…according to the Barcelona Clinic Liver Cancer staging system… Please add a reference.

Please define briefly the terms Peak and Ktrans the first time that they appear in the text.

IVIM-derived D and ADC values… ADC is not derived from the IVIM model. Please clarify better this point.

…The ADC ratio and D ratio… What do you mean by ADC and D ratio?

Methods

Please add a reference to the Response Evaluation Criteria in Solid Tumors (RECIST) 1.1.

Why 8 patients were excluded for no adequate hepatic lesion? What does this mean? Please clarify.

How many gradient directions were chosen for DWI?

Why ADC was calculated using multiple b-values of the IVIM sequence? Why DWI with b=1000 was not used for this aim? Why did you include the DWI sequence having 3 b-values? Please justify the use of multiple b-values for the calculation of ADC.

The equation : SIb1/SIb2 = e−(b1 − b2)ADC is suitable for 2 different b-values. Whereas you stated all b-values were used. Please check this.

… Interobserver variability were calculated… should be “was calculated”

Did you use a correction for multiple comparisons in your analysis? I think this must be added, if not possible you should explain why.

Results

In Figures 2 and 3 I do not see a direct match between DCE and, T1w and DWI/IVIM maps. I suggest co-register the images for better visualization.

Univariable analysis revealed… this should be “univariate analysis”

How parameters were classified as “High” or “low” in figure 4?

It would be nice to show some representative signal dynamics for both DCE and DWI/IVIM data.

How good was the data fitting?

Discussion

The authors must discuss with a higher level of detail why the ADC is correlated to some clinical outcomes whether D is not. Are D and ADC correlated? Please clarify this aspect

Furthermore, f is correlated to the overall survival but these results can be biased by the estimation errors that occur when a bi-exponential IVIM model is fitted. I recommend the author to introduce a further limitation of their work – the use of least-square fitting that can introduce biases for the estimation of f and D*- and state how this can be improved using recent Bayesian approaches such as – as examples - those described in the following papers:

  • Gustafsson, O., Montelius, M., Starck, G., & Ljungberg, M. (2018). Impact of prior distributions and central tendency measures on Bayesian intravoxel incoherent motion model fitting. Magnetic resonance in medicine, 79(3), 1674-1683.
  • Lanzarone, E., Mastropietro, A., Scalco, E., Vidiri, A., & Rizzo, G. (2020). A novel bayesian approach with conditional autoregressive specification for intravoxel incoherent motion diffusion‐weighted MRI. NMR in Biomedicine, 33(3), e4201.

Round 2

Reviewer 2 Report

I want to thank the authors for the work done on the manuscript. The paper was improved after the first review step and I think it is suitable for publication with few further changes.

I would suggest the authors not include into the abstract the results that are not significant after Bonferroni correction.

Author Response

Dear Editor and reviewer:

Thank you for the positive comments.

The readers may not understand the conclusion if ADC was removed due to nonsignificance after Bonferroni correction, so we add the description of Bonferroni correction in the abstract result to remind the readers the issues of multiple comparisons.

Moreover, Kep and slope were still significant after Bonferroni correction was performed (P < 0.005).

Thank you again for consideration of this manuscript.
